# Effectiveness of Disinfection with Chlorine Dioxide on Respiratory Transmitted, Enteric, and Bloodborne Viruses: A Narrative Synthesis

**DOI:** 10.3390/pathogens10081017

**Published:** 2021-08-12

**Authors:** Michele Totaro, Federica Badalucco, Anna Laura Costa, Benedetta Tuvo, Beatrice Casini, Gaetano Privitera, Giovanni Battista Menchini Fabris, Angelo Baggiani

**Affiliations:** 1Department of Translational Research and the New Technologies in Medicine and Surgery, University of Pisa, 56126 Pisa, Italy; micheleto@hotmail.it (M.T.); federica.b92@gmail.com (F.B.); anna.costa@med.unipi.it (A.L.C.); tuvobenedetta@hotmail.it (B.T.); beatrice.casini@med.unipi.it (B.C.); gaetano.privitera@med.unipi.it (G.P.); 2San Rossore Dental Unit, 56122 San Rossore, Italy; gbmenchinifabris@yahoo.it

**Keywords:** chlorine dioxide, SARS-CoV-2, enteric viruses, bloodborne viruses

## Abstract

A viral spread occurrence such as the SARS-CoV-2 pandemic has prompted the evaluation of different disinfectants suitable for a wide range of environmental matrices. Chlorine dioxide (ClO_2_) represents one of the most-used virucidal agents in different settings effective against both enveloped and nonenveloped viruses. This narrative synthesis is focused on the effectiveness of ClO_2_ applied in healthcare and community settings in order to eliminate respiratory transmitted, enteric, and bloodborne viruses. Influenza viruses were reduced by 99.9% by 0.5–1.0 mg/L of ClO_2_ in less than 5 min. Higher concentration (20 mg/L) eliminated SARS-CoV-2 from sewage. ClO_2_ concentrations from 0.2 to 1.0 mg/L ensured at least a 99% viral reduction of AD40, HAV, Coxsackie B5 virus, and other enteric viruses in less than 30 min. Considering bloodborne viruses, 30 mg/L of ClO_2_ can eliminate them in 5 min. Bloodborne viruses (HIV-1, HCV, and HBV) may be completely eliminated from medical devices and human fluids after a treatment with 30 mg/L of ClO_2_ for 30 min. In conclusion, ClO_2_ is a versatile virucidal agent suitable for different environmental matrices.

## 1. Introduction

Disinfection treatments in healthcare and community settings are aimed at achieving microbiological compliance for environmental matrices (drinking water, air, and surfaces). Disinfection is the final treatment after cleaning, which acts on the residual microbiological component, ensuring the absence of pathogens. Among the several chemical biocides, chlorine compounds such as chlorine dioxide, sodium hypochlorite, and chloramines are usually recognized as useful in reducing the infection risk related to environmental matrices in healthcare facilities [1].

The use of gaseous chlorine dioxide as a disinfection agent for drinking water has been increasing in recent years. It is an unstable gas produced on-site by mechanical generators using acid-based or electrolytic methods [1], and it is usually used for water treatment at a concentration between 0.1 and 5.0 mg/L.

It is used as an oxidant agent [2] decomposing the biofilm inside pipes and tanks [3], and it can react only by oxidation with a low trihalomethanes (THM) formation in water. In fact, it has an oxidizing effect on organic components originating from mainly oxidized byproducts and a small amount of chloro-organic compounds, while chlorine reacts with various substances via oxidation and electrophilic substitution [4]. Chlorine dioxide (ClO_2_) is more biocidal than both chlorine and chloramines, but it generates organoleptic defects in the water after treatment [5]. Actually, in comparison to chlorine, ClO_2_ reduces the generation of toxic halogenated disinfection byproducts, but ClO_2_ disadvantages are the formation of the organic halides chlorite/chlorate and the production of tastes and odors at concentrations of 0.5 mg/L [6]. 

Since 1940, data have been published demonstrating the chlorine dioxide action on bacteria, viruses, biofilms, protozoa, and algae [7]. In case of viruses, the inactivation mechanism differs from that of bacteria or other cells. The inactivation time of a virus is probably much shorter than that of a bacterium under the same conditions. ClO_2_ gas does not necessarily penetrate viruses to inactivate them. ClO_2_ reacts with one or more of the cysteine, tyrosine, and tryptophan amino acid residues of the spike proteins located on the enveloped viral surface. In the case of nonenveloped viruses, ClO_2_ acts on the viral genome [8,9,10].

The aim of this narrative review is the evaluation of the main studies performed on ClO_2_ activity on respiratory transmitted, enteric, and bloodborne viruses.

We provide an overview of the ClO_2_ features, usage settings, and its virucidal spectrum. The cited viruses were chosen on the basis of the available data and the references provided by some databases (PubMed, Scopus, and Google Scholar). 

A scientific study design with the research details is provided in Figure 1.

### 1.1. Virucidal Activity of ClO_2_

There are many studies on the virucidal activity of ClO_2_, involving both nonenveloped and enveloped viruses. Enveloped viruses are different from nonenveloped ones due to the presence of a lipid bilayer membrane outside the viral protein capsid, which contains proteins or glycoproteins. The presence of different functional groups on the outer surface of enveloped viruses impacts their survival in different environments [11,12]. A lot of factors have also been found to have a great impact on virus inactivation rates, such as the ClO_2_ dosage, time, pH, and temperature [13].

Sanekata suggested that enveloped viruses are inactivated more easily than nonenveloped viruses when exposed to 1.0 mg/L of ClO_2_ [14]. The disinfectant action on the enveloped proteins cause a failure of the viral attachment to the host cell, and so, the failure of cell invasion and infection [15]. About that, some authors carried out a study showing that ClO_2_ inactivates the virus thanks to its reaction with amino acids like cysteine and tryptophan [16]. Noss et Haunchman found that the site of the disinfectant action is a viral protein component; inactivating the coat viral protein inhibits the virus’s ability to attack the host cells [10].

These statements were supported by Olivieri et al., who showed that the viral proteins and lipids were sufficiently altered by ClO_2_ during a disinfection treatment [17]. Both the human and animal cells are not free from the ClO_2_ effects, but the damage was not comparable to that on microorganisms for several reasons. Not only are the mammalian cell sizes much larger and a greater disinfectant exposure is needed, but these cells have protection systems such as glutathione and further proteins [1,18].

### 1.2. ClO_2_ Activity on Respiratory Transmitted Viruses 

Sanekata et al. evaluated the antiviral activity of ClO_2_ and sodium hypochlorite against human influenza virus (IFV). ClO_2_ reduced influenza viruses by 99.9% at 1.0 ppm for 15 s, while the same antiviral activity was obtained for sodium hypochlorite at 100 ppm for 15 s [14].

Lenes studied the ClO_2_ action on the H1N1 virus. Tests were performed with contact times of 5 min and a residual ClO_2_ value of 0.5 mg/L. After the treatment, the H1N1 virus was not detected in any of the samples. The H1N1 virus is effectively inactivated by chlorine dioxide [19]. 

Ogata found that ClO_2_ antimicrobial activity is based on protein denaturation through amino acid residue oxidative modification. In particular, tyrosine and tryptophan residues in the hemagglutinin protein constitute an important active-site pocket for viral infectivity. ClO_2_ causes a functional modification of these amino acids with a loss of infectivity. Ogata demonstrated that ClO_2_ reacts with the tryptophan 153 residue, inactivating the hemagglutinin function [20,21,22]. 

Kalay-Kullay assessed that ClO_2_ action on amino acid residues could also be the mechanism for the antiviral activity on SARS-CoV. Spike proteins contain tyrosine, tryptophan, and cysteine residues with which the disinfectant can easily react in an aqueous solution with rapid virus inactivation [9]. Carducci et al. reported that SARS-CoV seeds in 100 mL of domestic sewage may be completely inactivated with a disinfection with 20 mg/L of ClO_2_ for 30 min [23]. Additionally, Wang performed a similar study with comparable results (10 mg/L of ClO_2_ for 10 min and 40 mg/L of ClO_2_ for 5 min, with total viral inactivation) [24].

New epidemiological approaches have been investigated in order to define wastewater surveillance, aimed at determining the spread, persistence, and detection of SARS-CoV-2 or other viruses in communities during their most critical epidemiological occurrences.

ClO_2_ in the most-used and effective method for the abatement of viral loads in these types of waters. It can inactivate SARS-CoVs completely after 30 min of exposure time and at a concentration of 40 mg/L [25].

The same result was not obtained after a sewage treatment with other disinfectants, especially if they were not chlorine-based.

Wastewater-based epidemiology is a new approach for monitoring viral pathogens spreading (SARS-CoV-2) in different contexts and countries. It is a valuable early warning system. It is a helpful alternative surveillance tool to subside the public health response, containing and measuring different infectious risk levels, mostly for poor sanitation settings [26].

Moreover, against the Measles virus, the ClO_2_ antiviral activity of 99.99% of the viral load was obtained at 10 mg/L for 30 s or at 100 mg/L for 15 s [14,27].

The principal tests performed on the respiratory transmitted viruses are summarized in Table 1.

### 1.3. ClO_2_ Activity on Enteric Viruses

The viral inactivation by ClO_2_ was studied by Thurston-Enriquez et al. on Adenovirus type 40 (AD40). The tests were performed in various conditions, such as different pH values (8 and 6) and temperatures values (15 °C and 5 °C). The ClO_2_ doses ranged from 0.67 to 1.28 mg/L.

The rate of AD40 inactivation was higher at pH 8 and/or a temperature of 15 °C than at pH 6 and/or a temperature of 5 °C. The ClO_2_ efficacy increased at higher experimental temperatures and pH levels. 

The concentration of ClO_2_ multiplied by the contact time with the virus needed for a 4-log inactivation (*Ct*_99.99%_) for AD40 at 5 °C was 1.28 and 0.67 mg/L × min at pH 6 and 8, respectively. The *Ct*_99.99%_ values for AD40 were about two times higher at pH 6 than at pH 8 [6].

A Chinese research group tested the dioxide chlorine action on the Human Hepatitis A Virus (HAV) at a concentration of 5 mg/L. After 60 min, they observed that the infectivity was not completely eliminated yet. Increasing the concentration to 7.5 mg/L, HAV was completely inactivated after 10 min. The action mechanism was found in the viral genome damage and/or viral proteins destruction. Li et al. assessed that the disinfectant damaged the 5′-nontranslated region (5′NTR) of the genome, blocking its replication and reacting with the viral proteins, stopping the interactions with the host cells [15]. The Department of Public Health of Parma reported a faster inactivation of HAV (only 30 s at a 0.8-mg/L concentration and 5 min at 0.4 mg/L) [15,28].

Li et al. revealed that ClO_2_ reduced the HAV infectivity through 5′-NCR damage [29]. Simonet et al. reported similar findings as that of Li et al. for Poliovirus-1 (PV-1), whereby the 5′-NCR and 3′-NCR of the PV1 genome appeared to be the most sensitive to the ClO_2_ treatment [30].

Several studies have been performed on the ClO_2_ actions against enterovirus such as the poliovirus, norovirus, and coxsackievirus. Brigano et al. [31] promoted a theory based on the thermodynamic analysis asserting that ClO_2_ inactivates viruses through the denaturation of protein coatings. Although ClO_2_ reacts with coated proteins, changing the pH value (pH 6), the critical target appears to be the viral RNA. Alvarez and O’Brien [32] observed that the poliovirus inactivation at pH 10.0 was faster than at pH 6.0 [33,34]. 

A possible explication may be related to the alkaline disinfectant dissociation in chlorite (ClO^2−^) and chlorate (ClO^3−^). Berman et al. observed the same effect for the Rotavirus. They compared the inactivation obtained by chlorine, ClO_2_, and monochloramine at 5 °C at pH 6 and 10 on a purified preparation of single virions with 0.5 mg/L of disinfectant. At pH 6, more than 4 log (99.99%) of the single virions were inactivated in less than 15 s with chlorine and monochloramine. With ClO_2_, this effect was observed at pH 10 [13].

For Poliovirus, Tenno Fujioka and Loh indicated that ClO_2_ damages the RNA, separating the RNA from the viral capsid at pH 10 but not at pH 6. Virion is converted into an empty capsid, suggesting that the loss of infectivity is due to a slight structural modification of the capsid [35]. The antiviral activity against Norovirus was tested by Sanekata using feline calicivirus (FCV). An initial antiviral effect was obtained with 1 ppm of ClO_2_ for 180 s, but the antiviral activity against >99.99% of the virus was obtained with 10 ppm of ClO_2_ for 15 s [14].

In a study conducted by Alvarez, FCV was completely eliminated in 30 min by ClO_2_ at a concentration of 0.2 mg/L, and Coxsackie B5 showed a similar behavior, being completely inactivated in 4 min with 0.4 mg/L of ClO_2_ and after 30 min at a 0.2 mg/L concentration [32].

Thurston-Enriquez et al. observed that the total inactivation could be achieved when Enterovirus 71 (EV71) was treated with a concentration of 0.5 mg/L ClO_2_ for over 30 min, 1.5 mg/L ClO_2_ for 25 min, or 2.0 mg/L ClO_2_ for 15 min. The inactivation was more effective at pH 8.2 than at pH 5.6 with 4.92 mg/L of ClO_2_ for 1 min. Considering the temperature parameter, the inactivation was faster at 36 °C than at 4 °C or 20 °C. The ClO_2_ efficacy for EV71 was pH and temperature-dependent. Similarly, the inactivation of AD40 and FCV by ClO_2_ was reported to be higher at 15 °C than at 5 °C [6]. Jin et al. evaluated the ClO_2_ effect on EV71 infectivity and the genomic integrity. With 13.51 mg/L of ClO_2_ for 1 min, the 5′-NCR was not amplified by RT-PCR, and the viral infectivity disappeared.

Harakeh tested the ClO_2_ effect on three viruses: Human rotavirus, Coxsackievirus B5, and Poliovirus 1. The three enteroviruses were tested at pH 7.2 and 15 °C. Coxsackievirus B5 was the most resistant, with 17.25 ppm needed, whereas 15 ppm of free residual ClO_2_ was required for a complete inactivation (99.99%) in 5 min [36].

The Poliovirus 1 genome was not affected after the treatment with 0.5 mg/L of ClO_2_ for 120 min. This was confirmed by Tenno et al., who observed that nearly all the Poliovirus 1 genome remained infectious after exposure to 0.2 mg/L of ClO_2_ for 30 min. The ClO_2_ dose relationship was clearly demonstrated by exposing the genome to a higher concentration of ClO_2_ (5 mg/L) for 30 min, which resulted in a significant degradation of the viral RNA.

Tachikawa et al. asserted that ClO_2_ in drinking water may reduce a 4-log Poliovirus load in 2.5 min at 20 °C and pH 7 [37,38,39].

The significant reduction of enteric virus outbreaks in developed countries is partly due to the ClO_2_ efficacy in waters and wastewaters; this is also true for the most resistant viruses, such as Poliovirus and other Picornaviridae. 

The principal tests performed on the enteric viruses are summarized in Table 2.

### 1.4. ClO_2_ Activity on Bloodborne Viruses

Farr et al. tested the ClO_2_ activity in Human immunodeficiency virus type 1 (HIV-1) inactivation. The virus was treated by adding 100 µL of ClO_2_ to 0.9 mL of HIV-1 stock for a final concentration of 30 ppm. The preparation was incubated at 25 °C for 5 min. To test the effect in human blood, 100 µL of ClO_2_ and 250 µL of human whole blood were added to 650 µL of HIV-1 stock in order to achieve viral inactivation. In the presence of medical supplies (plastic and paper materials), the virus was inactivated by adding 100 µL of ClO_2_ to 0.9 mL of HIV-1 stock. The results showed a viral reduction of 5.25 log. In the presence of blood or medical devices, the HIV-1 reduction was higher than 4 log [40].

A group of Japanese researchers carried out a study on HCV-positive (human hepatitis C) periodontitis patients in order to evaluate the viral elimination on the ultrasonic cleaning device after a treatment with ClO_2_. 

After the periodontal activity, the instrument was disinfected with 0.02% ClO_2_ for 10 min. The total absence of HCV genomic units was assessed in a RT-PCR test directly performed on the devices [41]. Another study about the surgical instruments was performed by Isomoto et al. on endoscope disinfection after procedures on HCV-positive patients. The devices were reprocessed with 30 mg/L of ClO_2_ for 5 min. HCV genomic units were not detected in the RT-PCR test after the treatment [42].

Aseptrol^®^, containing a noncorrosive formula of ClO_2_, may eliminate the Hepatitis B virus (HBV) genome if it is used at 24 mg/L for 5 min on environmental surfaces [43]. 

The principal tests performed on the enteric viruses are summarized in Table 3.

## 2. Conclusions

Chlorine dioxide has been widely applied in environmental matrix disinfections, mostly for waters and wastewaters. In healthcare settings, disinfection procedures have to ensure viral inactivation in order to prevent outbreaks and epidemic occurrences. The use of ClO_2_ is the final step in a virucidal agent [44]. In 2018, the United Stated Environmental Protection Agency (USEPA) [45] cited adenoviruses, caliciviruses, enteroviruses, and hepatitis A virus as microbiological drinking water contaminants that may be disseminated through aquatic environments. The evaluation of the efficacy of alternative disinfectants such as ClO_2_ is important to perform in order to eliminate these virus species from drinking water. There have been many studies reporting virucidal activity against waterborne and nonwaterborne viruses, including nonenveloped viruses (e.g., adenoviruses and enteroviruses) and enveloped viruses (e.g., influenza viruses). 

Considering enteric viruses, ClO_2_ concentrations ranging from 0.2 to 1.0 mg/L ensure at least a 99% viral reduction of AD40, HAV, Coxsackie B5 virus, Norovirus, Rotavirus, Feline Calicivirus, and EV71 in less than 30 min. Moreover, ClO_2_ is the most-used disinfectant for drinking water treatment, and its efficacy may be achieved by the routine of continuous water chlorination, as described elsewhere (0.2 mg/L of chlorine dioxide) [46,47].

Similar results have been obtained for respiratory transmitted virus elimination. In particular, 0.5–1.0 mg/L of ClO_2_ for less than 5 min can reduce influenza viruses by 99.9% (the H1N1 virus included). Sewage and wastewaters may be treated with 20 mg/L of ClO_2_ in order to obtain a total SARS-CoV-2 elimination. At last, considering bloodborne viruses, 30 mg/L of ClO_2_ eliminates, in 5 min, HIV-1 in human organic fluids (blood) or medical supplies. HCV may be eliminated from medical devices after an application of 30 mg/L of ClO_2_ for 10 min. This narrative synthesis highlights the need for disinfection procedures with ClO_2_ applied to different matrices. The versatility of ClO_2_ is widely known for disinfection procedures in large and populated indoor environments, such as hospitals and generic healthcare facilities. Moreover, considering the high vulnerability of infected patients hosted in infectious disease wards and intensive care units, the application of ClO_2_ may be the solution for routine disinfection protocols in order to minimize the risk of viral agent transmissions (respiratory transmitted, enteric, or bloodborne viruses) and nosocomial and community infections [48,49]. The international recommendations published for SARS-CoV-2 prevention in hospital settings described the importance of environmental sanitization using only biocides tested for virucidal activity, as required by UNI EN 14476:2019 [50]. The Italian Institute of Health included ClO_2_ as a possible disinfectant for hospital sanitization in rooms having a high infectious risk [51].

This narrative synthesis aimed to highlight the versatility and suitability of ClO_2_ in different settings (mostly for waters) and against different viral agents.

It may be clear that ClO_2_ is not a therapeutic product. It cannot be used for healing human or animal tissues, with the exception of skin and mucosa antisepsis.

In conclusion, ClO_2_ is one of the most-used biocides for different environmental settings. It is a recognized versatile virucidal agent, and its efficacy is dose-dependent. It may be used as a high-, intermediate-, or low-level agent [48,52], in accordance with the suitable disinfection grade needed in different situations.

## Figures and Tables

**Figure 1 pathogens-10-01017-f001:**
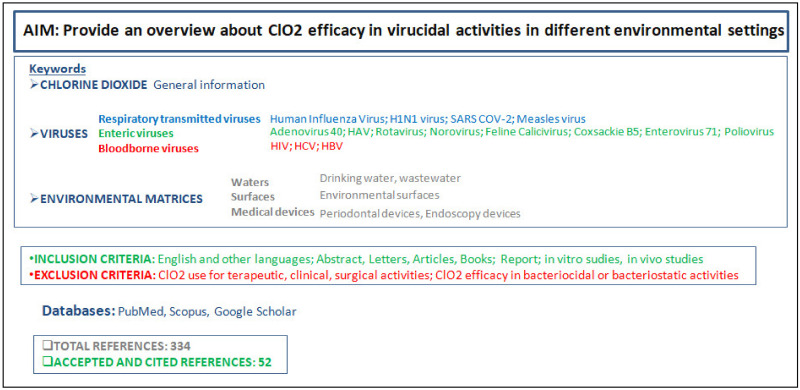
Study design representation: aim, keywords, inclusion and exclusion criteria, and references.

**Table 1 pathogens-10-01017-t001:** Literature data on the reduction of respiratory transmitted viruses following chlorine dioxide treatments at different times and concentrations.

Chlorine Dioxide Concentration	Exposure Time	Viral Reduction	Author and Year
1 mg/L	15 s	Human influenza virus (99.9%)	Sanekata 2010
0.5 mg/L	5 min	H1N1 virus (99.9%)	Lénès 2010
20 mg/L in sewage	30 min	SARS-CoV-2 (99.99%)	Carducci 2020
10 mg/L 100 mg/L	30 s 15 s	Measles virus (99.99%)	Sanekata 2010

**Table 2 pathogens-10-01017-t002:** Literature data on the reduction of enteric viruses following chlorine dioxide treatments at different times and concentrations.

Chlorine Dioxide Concentration	Exposure Time	Viral Reduction	Author and Year
0.67 mg/L (pH 8) 1.28 mg/L (pH 6)	1 min	Adenovirus type 40 (4 log)	Thurston-Enriquez 2005
7.5 mg/L	10 min	Human Hepatitis A Virus (100%, total genomic damage)	Li 2004
0.8 mg/L	0.5 min	Human Hepatitis A Virus (99%)	Zoni 2007
0.4 mg/L	5 min	Human Hepatitis A Virus (99%)	Zoni 2007
0.5 mg/L	15 s	Rotavirus (4log)	Berman 1984
1 mg/L 10 mg/L	3 min 15 s	Norovirus (99.99%) Norovirus (99.99%)	Sanekata 2010
0.2 mg/L	30 min	Feline Calicivirus (99.99%)	Zoni 2007
0.4 mg/L	4 min	Coxsackie B5 virus (99.99%)	Zoni 2007
0.5 mg/L	30 min	Enterovirus 71 (99.99%)	Thurston-Enriquez 2005
0.5 mg/L	30 min	Poliovirus 1 (RNA damage)	Simonet 2005
1 mg/L	2.5 min	Poliovirus (4 log)	Tachikawa 1993

**Table 3 pathogens-10-01017-t003:** Literature data on the reduction of bloodborne viruses following chlorine dioxide treatments at different times and concentrations.

Chlorine Dioxide Concentration	Exposure Time	Viral Reduction	Author and Year
30 mg/L (in blood sample)	5 min	Human Immunodeficiency Virus type 1 (4 log)	Farr 1993
0.02% 30 mg/L	10 min 5 min	Hepatitis C Virus (periodontal device) (99.99%) Hepatitis C Virus (endoscopy device) (99.99%)	Watamoto 2013 Isomoto 1998
24 mg/L (Aseptrol^®^)	5 min	Hepatitis B Virus (99.99%)	Ebonwu 2010

## Data Availability

The data presented in this study are openly available.

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
