# Peer review of "Effectiveness of Disinfection with Chlorine Dioxide on Respiratory Transmitted, Enteric, and Bloodborne Viruses: A Narrative Synthesis"

_pathogens, 2021, doi:10.3390/pathogens10081017_

Round 1
Reviewer 1 Report
This review is well written and organized and deals an interesting and actual topic. The bibliography cited is appropriate and reports the main studies on the application of ClO2 (mainly as a disinfectant in the treatment of water and wastewater). Check only the error in line 139.
Author Response
REVIEWER 1
This review is well written and organized and deals an interesting and actual topic. The bibliography cited is appropriate and reports the main studies on the application of ClO2 (mainly as a disinfectant in the treatment of water and wastewater). Check only the error in line 139.
Thank you for your comment. Line 139 has been edited
Reviewer 2 Report
This manuscript lacks the description of a scientific study design and can not sure to present the complete information related to the study topic. In general, it provide limited information and the effectiveness of disinfection with chlorine dioxide can not covered all the respiratory, enteric, and blood-borne viruses as well.Author Response
REVIEWER 2
This manuscript lacks the description of a scientific study design and can not sure to present the complete information related to the study topic. In general, it provide limited information and the effectiveness of disinfection with chlorine dioxide can not covered all the respiratory, enteric, and blood-borne viruses as well.
Thank you for your opinion.
A scientific study design scheme has been added in order to give a more accurate information related the purpose, research details, keywords, inclusion and exclusion criteria.
Manuscript has been revised in order to give more information about the broad-spectrum antiviral activity of ClO2.
However the cited viruses are chosen on the basis of the available data and references provided by common scientific databases (pubmed, scopus, google scholar).
More data about further viruses have been added in order to give a more complete vision of the ClO2 virucidal activity.
In detail, we added data about virucidal activities of ClO2 on Poliovirus, Measles virus, HBV.
In conclusion section more information about the role of the narrative synthesis has been added, as suggested by Reviewer 3.
Reviewer 3 Report
It is clear that disinfection processes are quite important especially in a series of situations, such as healthcare settings where materials and surfaces may contact many pathogens. It is also important to control and avoid pathogen transmission via fomites. In this context, this review is focused in the use of chlorine dioxide as a virucidal agent. It reviews the published information on that issue. Some issues could be clarified and improved.
Specific comments:
- Please note that ClO2 has become a highly sensitive issue during in the pandemic, as there were many misleading informations, especially in social media, recommending people the use of this compound as a therapeutic agent. Therefore, this reviewer thinks that it may be extremely important to clearly state in the manuscript text that virucidal/disinfectant activity of ClO2 does not mean at all antiviral activity and, therefore, it is not a therapeutic agent for humans or animals.
In line with previous comment, lines 100-106 would be eliminated from the manuscript, as they do not refer to disinfectant activity. This refers to a unique publication with controversial results.
- An issue not addressed in this manuscript that would be interesting: epidemiological surveillance through analysis of wastewaters has been demonstrated as a very good measure to anticipate virus outbreaks. Since ClO2 is used in wastewaters for disinfection, could it affect virus detection? Is there any comparison in virus detection, for epidemiological surveillance, in treated vs. non-treated wastewaters?
- As clearly indicated throughout the manuscript, it is known for years that ClO2 is a suitable disinfectant eliminating different pathogens including viruses. Then, which is the aim of this review? Based on the compiled information, is there any criticism and/or recommendation from the authors that may help the scientific community, clinicians, etc?
Minor comments:
- Line 71. Reference #17 may not be adequate here, as it is not on antiviral activity but on antibacterial activity.
- Line 139. There is a strikethrough “the”
Author Response
REVIEWER 3
Please note that ClO2 has become a highly sensitive issue during in the pandemic, as there were many misleading informations, especially in social media, recommending people the use of this compound as a therapeutic agent. Therefore, this reviewer thinks that it may be extremely important to clearly state in the manuscript text that virucidal/disinfectant activity of ClO2 does not mean at all antiviral activity and, therefore, it is not a therapeutic agent for humans or animals.In line with previous comment, lines 100-106 would be eliminated from the manuscript, as they do not refer to disinfectant activity. This refers to a unique publication with controversial results.
We agree. Statement has been removed avoiding some doubts about the possible use of ClO2 as drug.
An issue not addressed in this manuscript that would be interesting: epidemiological surveillance through analysis of wastewaters has been demonstrated as a very good measure to anticipate virus outbreaks. Since ClO2 is used in wastewaters for disinfection, could it affect virus detection? Is there any comparison in virus detection, for epidemiological surveillance, in treated vs. non-treated wastewaters?
A reflection about this issue has been added in the text, mostly for SARS COV-2 risk in wastewaters.
As clearly indicated throughout the manuscript, it is known for years that ClO2 is a suitable disinfectant eliminating different pathogens including viruses. Then, which is the aim of this review? Based on the compiled information, is there any criticism and/or recommendation from the authors that may help the scientific community, clinicians, etc?
Aim of study is the need to highlight the potential activity, the versatility and the large use of one of the most worldwide used disinfectant.
Further statements have been added in conclusion section.
In the section we state that ClO2 is a good compound but it may not be uses as a therapeutic agent, as often described by mass media or social networks.
Minor comments:
- Line 71. Reference #17 may not be adequate here, as it is not on antiviral activity but on antibacterial activity.
Reference has been removed
- Line 139. There is a strikethrough “the”
It has been removed
Round 2
Reviewer 2 Report
None
Reviewer 3 Report
The manuscript has been improved following the reviewers' suggestions.